# Design and Application of MEMS-Based Hall Sensor Array for Magnetic Field Mapping

**DOI:** 10.3390/mi12030299

**Published:** 2021-03-12

**Authors:** Chia-Yen Lee, Yu-Ying Lin, Chung-Kang Kuo, Lung-Ming Fu

**Affiliations:** 1Institute of Materials Engineering, National Pingtung University of Science and Technology, Pingtung 912, Taiwan; leecy@mail.npust.edu.tw (C.-Y.L.); oway151@gmail.com (C.-K.K.); 2Department of Materials Engineering, National Pingtung University of Science and Technology, Pingtung 912, Taiwan; capricorn15286@gmail.com; 3Department of Engineering Science, National Cheng Kung University, Tainan 701, Taiwan

**Keywords:** hall sensor, hall effect, ion implantation, MEMS, sensor array

## Abstract

A magnetic field measurement system based on an array of Hall sensors is proposed. The sensors are fabricated using conventional microelectromechanical systems (MEMS) techniques and consist of a P-type silicon substrate, a silicon dioxide isolation layer, a phosphide-doped cross-shaped detection zone, and gold signal leads. When placed within a magnetic field, the interaction between the local magnetic field produced by the working current and the external magnetic field generates a measurable Hall voltage from which the strength of the external magnetic field is then derived. Four Hall sensors are fabricated incorporating cross-shaped detection zones with an identical aspect ratio (2.625) but different sizes (S, M, L, and XL). For a given working current, the sensitivities and response times of the four devices are found to be almost the same. However, the offset voltage increases with the increasing size of the detection zone. A 3 × 3 array of sensors is assembled into a 3D-printed frame and used to determine the magnetic field distributions of a single magnet and a group of three magnets, respectively. The results show that the constructed 2D magnetic field contour maps accurately reproduce both the locations of the individual magnets and the distributions of the magnetic fields around them.

## 1. Introduction

Hall sensors based on CMOS (Complementary Metal-Oxide-Semiconductor Transistor) technology are widely applied in the manufacturing, medical devices, consumer electronics, automobile, and aerospace fields nowadays due to their low cost, high integration ability, and good reliability [1,2,3,4,5,6,7]. Hall sensors have many practical advantages, including non-contact operation, high linearity, physical sturdiness, and versatility [8,9,10]. As a result, they have attracted significant attention throughout industry and academia in recent years [11,12,13]. In the last decade, many sensor fusion techniques have been developed to improve sensing field mapping [14,15]. Such systems could be used for condition monitoring and prognosis of machines [16,17].

Lozanova et al. [18] fabricated a Hall magnetic sensor consisting of two parallel-field Hall devices for obtaining in-plane magnetic field measurements and one orthogonal Hall element for acquiring out-of-plane (i.e., vertical) magnetic field measurements. It was shown that six contacts were sufficient to acquire simultaneous measurements of the three orthogonal magnetic field components. In a later study [19], the same group presented a single-chip device based on a rectangular n-type silicon substrate for measuring two orthogonal magnetic-field components using a common transduction zone and just four contacts. The lateral sensitivity and vertical sensitivity of the proposed device were shown to be S_x_ = 17 V/AT and S_z_ = 23.3 V/AT, respectively. Furthermore, the channel cross-talk at an induction of B ≤ 1.0 T was found to be no more than 3%. Zhao et al. [20] integrated six pairs of permanent magnets and six Hall sensors to realize a six-degree-of-freedom (6-DOF) measurement system for a precision positioning stage. The experimental results showed that the proposed system enabled the in-plane stage displacement to be controlled to within 0.23 mm and the angular displacement to be controlled within 0.07°. Xu et al. [21] demonstrated the potential for realizing Hall element sensors based on graphene rather than silicon substrates. It was shown that the higher carrier sensitivity and atomically thin active-body of graphene made possible the realization of a magnetic sensor with high sensitivity, excellent linearity, and outstanding thermal stability. Jones et al. [22] developed a Hall effect tactile sensor for hand splinting applications. The design parameters, mechanical response, and force range of the proposed device were optimized by finite element simulations. The results obtained using a prototype device showed that the optimized design achieved a pressure range of 45 kPa in the normal direction and 6 kPa in the shear direction. Berus et al. [23] prepared Hall sensors capable of working over a broad range of temperatures based on heavily-doped n-InSb epitaxial thin films. The proposed sensors exhibited a temperature coefficient of the magnetic sensitivity of less than 0.01% per degree and were virtually independent of the temperature up to 400 K. Uzlu et al. [24] fabricated a gate-tunable graphene-based Hall sensor on a flexible polyimide (PI) substrate. In the proposed device, the signal-to-noise ratio was improved through the use of an AC-modulated gate electrode, which increased the sensitivity of the device and reduced the off-set voltage compared to a traditional Hall sensor with a static operation.

As shown in Figure 1, when a Hall sensor is placed perpendicularly within an external magnetic field, the current flowing through the sensor is deflected toward one side of the substrate and creates an orthogonal voltage (referred to as the Hall voltage) with a magnitude proportional to both the working current and the strength of the external magnetic field. In physics, the Lorentz force is the magnetic force acting on a point charge in the presence of an electromagnetic field. In particular, a particle of charge q moving with velocity v in an electric field E and magnetic field B experiences a force of:F = q (E + v × B)(1)

In other words, the electromagnetic force acting on charge q is a combination of a force in the direction of the electric field E proportional to the magnitude of the field and the quantity of the charge, and a force acting on the charge at right angles to the magnetic field B and charge flow direction proportional to the magnitude of the electric field, the charge, and the velocity. Similar formulae can be used to describe the magnetic force acting on a current-carrying wire or wire loop moving through a magnetic field.

Generally speaking, Hall sensors made from III–V semiconductor materials, such as InSb tend to outperform those based on silicon. However, such devices are more expensive and difficult to fabricate than silicon devices. Moreover, they are less easily integrated with the signal-processing circuits and functions required to carry out the sensing operation [11]. Accordingly, the present study fabricates a magnetic field measurement system based on an array of Hall sensors constructed on silicon substrates using simple microelectromechanical systems (MEMS) technologies. Four Hall sensors incorporating detection zones with an identical aspect ratio but different sizes (S, M, L, and XL) are designed, manufactured, and characterized. Finally, a 3 × 3 Hall sensor array is assembled into a 3D-printed frame to measure the magnetic field mapping of both a single magnet and an arrangement of three magnets, which will be an important tool in the Nondestructive-Testing (NDT) field.

## 2. Principle and Design

When a current passing through a semiconductor material flows in a direction perpendicular to that of an external magnetic field, the carriers in the semiconductor are deflected to one side and produce a potential difference called the Hall voltage with a magnitude equal to:(2)$$V_{H}={I\;\times \;B\;\times \;R}_{H}∕d$$
where I is the electrical current, B is the magnetic field strength, d is the thickness of the semiconductor material and $R_{H}=r_{H}∕\mathrm{nq}$ is the so-called Hall resistivity, in which n is the density of the charge carriers, q is the carrier charge, and $r_{H}$ is the Hall scattering factor depending on the used semiconductor material and the dominant charge carrier mechanism.

Figure 2 presents a schematic illustration of the sensor proposed in the present study consisting of a 4-inch silicon wafer, a silicon dioxide insulating layer, metal leads, and a cross-shaped phosphide-doped detection zone [25,26,27] For comparison purposes, four devices are fabricated, in which the cross-shaped detection zones have the same aspect ratio (L/W = 2.625), but different sizes, namely S, M, L, and XL. In the proposed devices, the analog signals are amplified by a large-current operational amplifier (AD620) and then supplied to an analog-to-digital converter (MEGA2560). Utilizing interpolation elements to smoothen planar display, the digital signals are then processed in LabVIEW to determine the corresponding magnetic field strength and distribution (Figure 3).

## 3. Fabrication

Figure 4 presents schematic illustrations showing the basic steps of the sensor fabrication process. At first, a silicon oxide isolation layer with a thickness of 0.5 μm was deposited on the surface of the 4-inch P-type < 100 > Si wafer (Thickness: 525 +/ −25 µm, Internal Resistivity: 1–20 Ω-cm) using a High-Density Plasma Chemical Vapor Deposition (HDPCVD) technique (Figure 4a) [28,29]. A photolithography method was employed to pattern (Figure 4b) and etch the cross-shaped detection zone in the Si substrate (Figure 4c) and the detection zone was then implanted with phosphide ions.

(Doping energy: 100 KeV, Depth: 0.1 µm, Concentration: 10^15^ ions/cm^2^) (Figure 4d). The residual photoresist (PR) was then stripped away in acetone solution and an annealing process was performed at 900 °C for 1 min(Figure 4e). Apparently, a thin layer of silicon oxide was oxidized on the detection area. A PR layer was then deposited on the surface of the silicon wafer using the photolithography method [30,31] to pattern (Figure 4f) and etch the contact windows of the leads (Figure 4g) on each branch of the cross-shaped detection zone (Figure 4h). Finally, the Electron Beam Evaporation (EBE) method was used to pattern the sensor surface with Au/Cr leads for electrical connection purposes (Figure 4i–k).

Figure 5a presents a photograph of the complete Hall sensors. The sensors were attached to a glass fiberboard using UV glue and encapsulated in resin. Finally, 2 × 2 pin headers were bonded to the sensors using conductive silver glue to form the final sensor assembly Figure 5b.

## 4. Results and Discussion

### 4.1. Sensitivity Test

Figure 6 illustrates the experimental setup used in the present study to characterize and compare the four Hall sensors with different detection zone dimensions. The experiments commenced by measuring the voltage response of the four sensors given different working currents (2, 7, and 10 mA) and magnetic field strengths in the range of 0–5200 Gauss. The corresponding results are presented in Figure 7. As expected, the measured Hall voltages of each sensor increase with the increasing working current. Moreover, for a given working current, the Hall voltage increases linearly with an increasing magnetic field strength (R^2^ = 0.9955–0.9982). Finally, for a constant magnetic field strength and working current, the Hall voltage increases with an increasing size of the detection zone.

Table 1 compares the sensitivities of the four sizes of devices at each of the considered working currents. It is seen that even though the detection zones in the different devices have different sizes, the devices have the same sensitivity since they conduct the same current in every case. The offset voltage increases with an increasing size of the detection zone at different identical working currents. In other words, the surface electrical properties of the detection zone are more susceptible to the effects of the mechanical stress caused by the manufacturing and packaging processes as the size of the detection zone increases.

### 4.2. Hysteresis Test

Hysteresis is the time-based dependence of the response on the current and past state of the input. Specifically, the output of the Hall sensor depends not only on the instantaneity of the input but also on its history. If the design is not good enough in the early stages, the hysteresis may make the measurement results different from the previous results [32]. To realize the overall characteristics of the proposed Hall sensor, it is necessary to observe the hysteresis phenomenon under different magnetic fields. The experimental results (Figure 8) show the voltage path used to increase and decrease the magnetic field when the operating current of the Hall sensor of size S is 10 mA. It can be seen that as the magnetic field is less than 4000 Gauss, the two paths are actually almost superimposed. When the magnetic field is greater than 4000 Gauss, the maximum observed hysteresis is 170 Gauss (4.25%). This phenomenon greatly reduces the deviation of the magnetic field measurement, especially at a magnetic field strength less than 4000 Gauss.

### 4.3. Temperature Effect Test

One of the key characteristics of the Hall sensor is the effect of offset voltage and sensitivity on temperature. With the advancement of fabrication technology, different offset compensation methods are known, such as calibration, device symmetry, mutual compensation, and reduction of rotating current offset [33,34]. Figure 9 shows the relationship between the offset voltage, sensitivity, and temperature of the S-size Hall sensor when the working current is 10 mA. It can be found that due to the overall temperature coefficient of resistance (TCR) when the ambient temperature is increased from 30 °C to 50 °C at a rate of 0.1 mV/°C, the offset voltage of the Hall sensor increases linearly, and the sensitivity of the sensor also increases with the increase in temperature at a rate of 0.01 µV/Gauss-°C. It is obvious that as the environmental temperature exceeds 50 °C, the offset voltage and sensitivity of the Hall sensor greatly increase due to the greater influence of temperature on the sensor. Temperature compensation can be incorporated according to the results of the temperature effect test for accurate measurement.

### 4.4. Time Response Test

The responsivity of the four sizes of Hall sensors was evaluated by covering the sensors with a non-metal plate and placing them within a magnetic field with a strength of 2000 Gauss. The plate was abruptly removed from the Hall sensors and the corresponded Hall voltages were recorded. In Figure 10, it is seen that the response time is 8 ms for all four sizes of devices as the working current was 10 mA.

### 4.5. Integration Test

Having characterized the sensing performance of the proposed Hall sensors, nine sensors were assembled into a 3D-printed plastic frame to create a 3 × 3 sensing array (Figure 11). The size of utilized sensors is S, and the working current is 10 mA. The feasibility of the sensing array was investigated by detecting the magnetic field distributions of a single magnet and a group of three magnets attached to the underside of an iron plate, respectively (Figure 12a,b). The magnetic field strength of each circle magnet (Diameter: 18 mm, Thickness: 5 mm) is 1420 Gauss. The detection results are presented in Figure 12c,d respectively. It is seen that for both arrangements of the magnets, the 2D magnetic field contour maps constructed in LabView accurately detect both the locations of the individual magnets and the distributions of the magnetic fields around them. The operational video is attached.

## 5. Conclusions

This study has designed and fabricated Hall sensors with application to magnetic field strength measurement. On a P-type silicon substrate, the device features a phosphide-doped cross-shaped detection zone and is fabricated using simple MEMS-based techniques, including HDPCVD sputtering, photolithography patterning, ion implantation, and EBE metal lead deposition. Four kinds of devices have been constructed with detection zones of identical aspect ratio (2.625) but different sizes (S, M, L, and XL). The experimental results have shown that all four sizes of devices have a linear response over the range of 0–5200 Gauss for working currents of 2 mA, 7 mA, and 10 mA. For a constant working current, the four sensors have an identical sensitivity and a similar response time (8 ms) despite the difference in the size of the detection zone. However, the offset voltage increases with an increasing detection zone size; indicating that the surface electrical properties of the detection zone are more susceptible to the effects of the manufacturing and packaging processes as the size of the detection zone increases. The practical feasibility of the proposed sensors has been demonstrated by measuring the magnetic field mapping of a single magnet and a group of three magnets, respectively, using a 3 × 3 array of sensors assembled into a 3D-printed frame, which will be an important tool in the NDT field.

## Figures and Tables

**Figure 1 micromachines-12-00299-f001:**
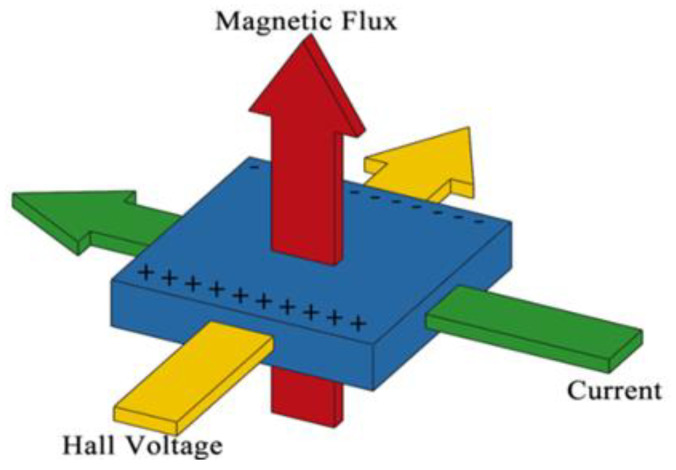
Schematic illustration of Hall effect.

**Figure 2 micromachines-12-00299-f002:**
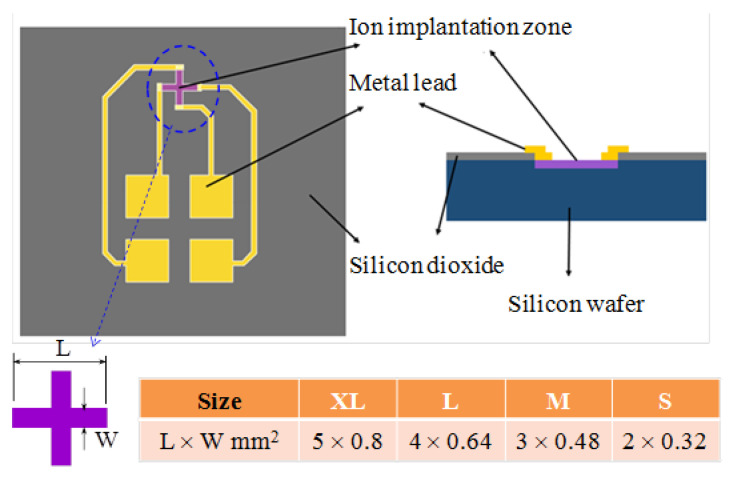
Schematic illustration of Hall sensor design and dimensions of four sizes of cross-shaped detection zones.

**Figure 3 micromachines-12-00299-f003:**
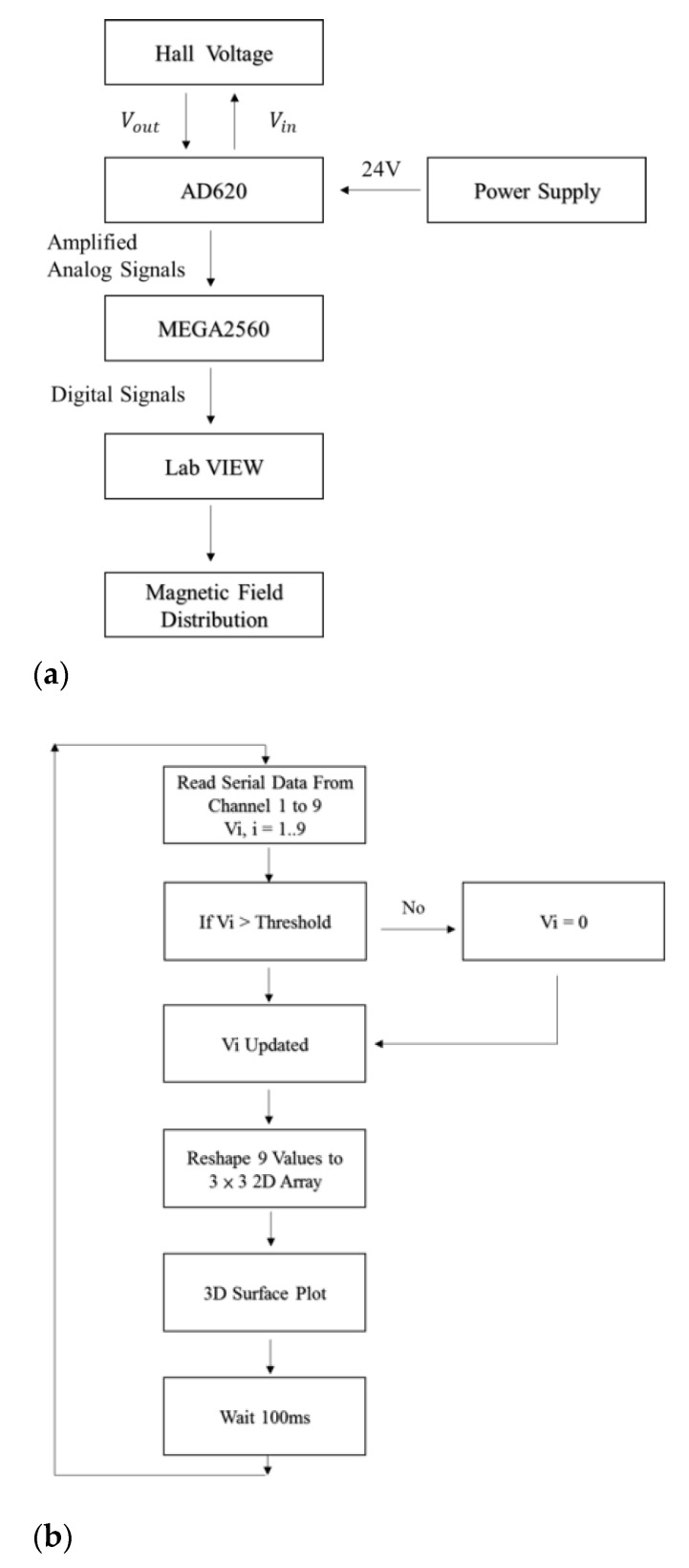
Block diagrams showing (**a**) analog circuit and signal conversion path of proposed Hall sensor array and (**b**) LabVIEW program of the visualization platform of the magnetic field mapping.

**Figure 4 micromachines-12-00299-f004:**
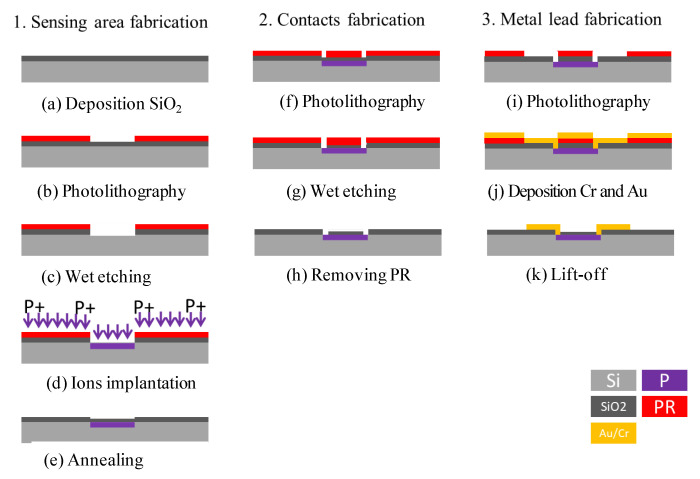
Schematic illustration showing the main steps of the fabrication process of Hall sensor: (**a**) deposition of SiO_2_, (**b**) photolithography, (**c**) wet etching, (**d**) ion implantation, (**e**) annealing, (**f**) photolithography, (**g**) wet etching, (**h**) removing PR, (**i**) photolithography, (**j**) deposition of Cr and Au, and (**k**) lift-off.

**Figure 5 micromachines-12-00299-f005:**
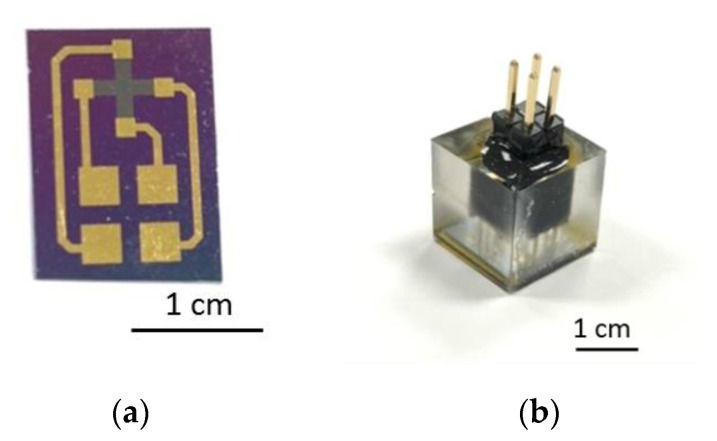
Photographs of: (**a**) complete Hall sensor, and (**b**) resin-encapsulation of individual Hall sensor.

**Figure 6 micromachines-12-00299-f006:**
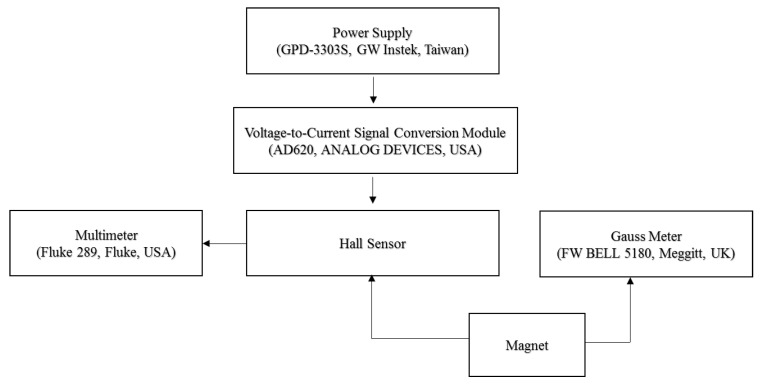
Experimental setup.

**Figure 7 micromachines-12-00299-f007:**
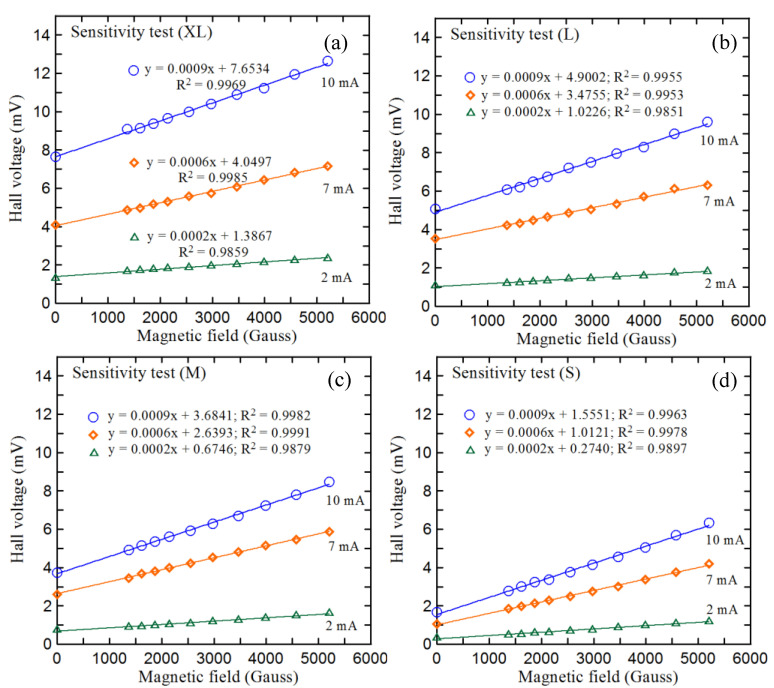
Sensitivity results for Hall sensors of size (**a**) XL, (**b**) L, (**c**) M, and (**d**) S.

**Figure 8 micromachines-12-00299-f008:**
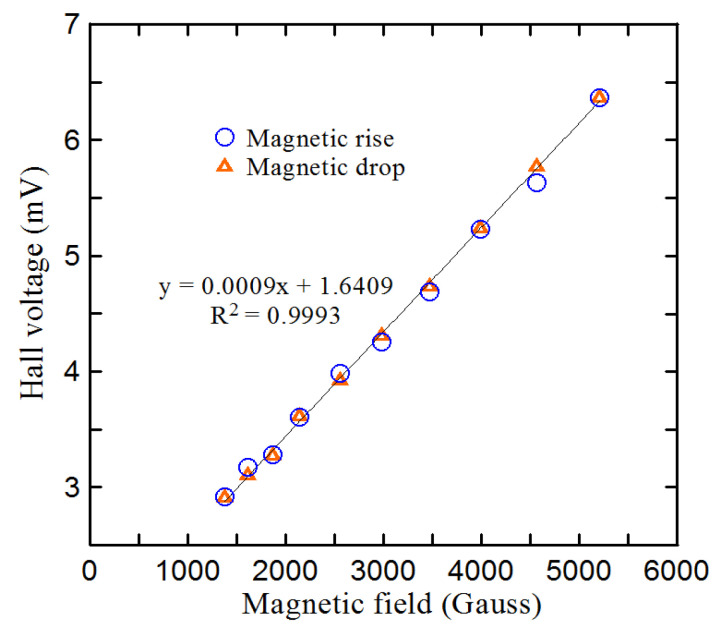
Magnetic hysteresis curve for the Hall sensor of size S at 10 mA of working current.

**Figure 9 micromachines-12-00299-f009:**
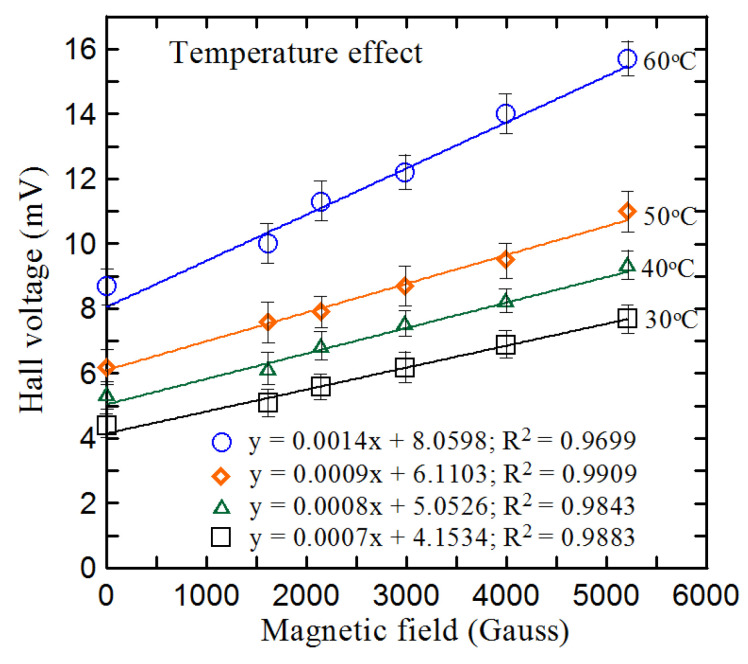
Temperature effect on voltage offset and sensitivity of the S-sized hall sensors at 30 °C, 40 °C, 50 °C, and 60 °C as the working current is 10 mA.

**Figure 10 micromachines-12-00299-f010:**
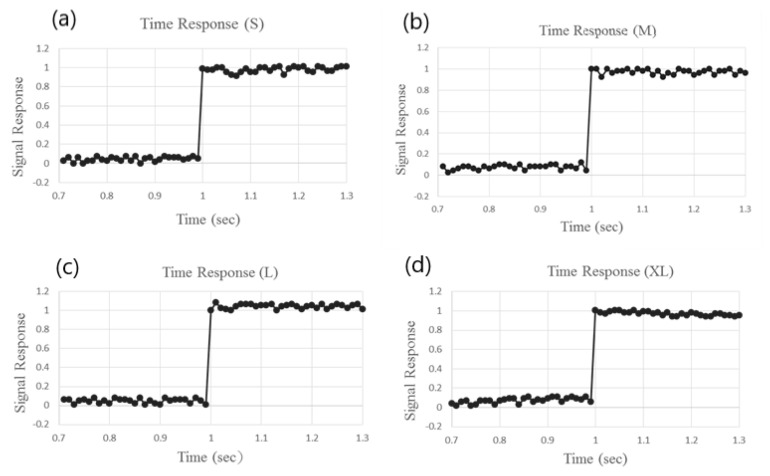
Time responses of Hall sensors of various sizes: (**a**) S, (**b**) M, (**c**) L, and (**d**) XL.

**Figure 11 micromachines-12-00299-f011:**
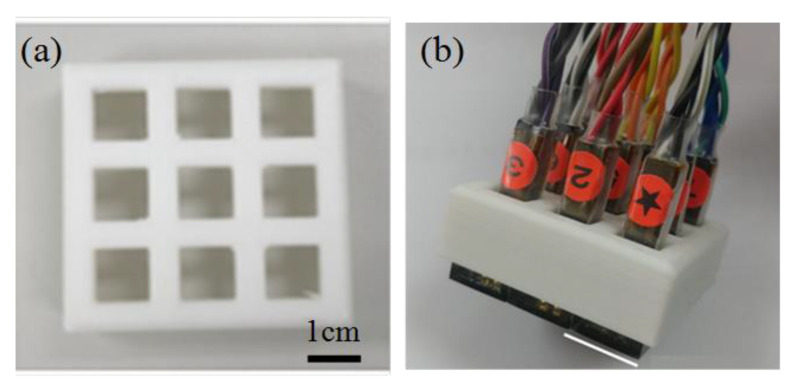
Photographs of: (**a**) 3D-printed Hall sensor frame, and (**b**) 3 × 3 Hall sensor array consisting of 9 sensing modules.

**Figure 12 micromachines-12-00299-f012:**
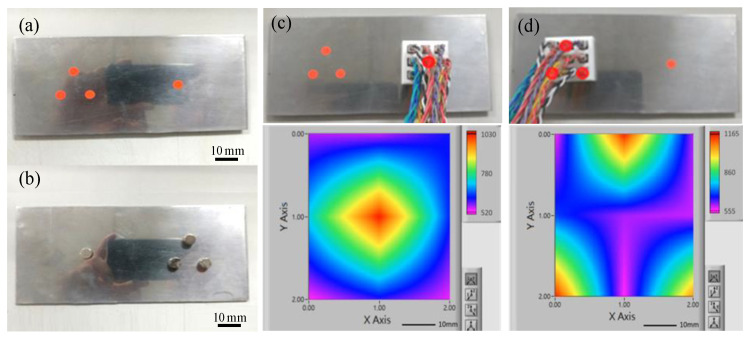
Photographs of: (**a**,**b**) front and rear sides of the iron plate with attached magnets; (**c**,**d**) single magnet and three-magnet detection arrangements and corresponded 2D magnetic field distributions (unit: Gauss).

**Table 1 micromachines-12-00299-t001:** Offset voltage, sensitivity, and coefficients of determination of four sizes of Hall sensors under three different working currents.

		XL	L	M	S
working current 10 mA	offset voltage (mV)	7.640	5.063	3.744	1.668
	sensitivity (μV/Gauss)	0.923	0.914	0.930	0.909
	R-square	0.9969	0.9955	0.9982	0.9963
working current 7 mA	offset voltage (mV)	4.083	3.541	2.613	1.056
	sensitivity (μV/Gauss)	0.615	0.621	0.598	0.607
	R-square	0.9985	0.9953	0.9991	0.9978
working current 2 mA	offset voltage (mV)	1.302	1.069	0.737	0.330
	sensitivity (μV/Gauss)	0.214	0.196	0.231	0.208
	R-square	0.9859	0.9851	0.9879	0.9897

## Data Availability

The data presented in this study are available on request from the corresponding author.

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
