# Peer review of "Design and Application of MEMS-Based Hall Sensor Array for Magnetic Field Mapping"

_micromachines, 2021, doi:10.3390/mi12030299_

Round 1

Reviewer 1 Report

The paper proposes an interesting approach to sensor fusion in magnetic fields, but it requires of a number of amendments prior to publication.

  • Generic sensor fusion techniques should be referenced. On the other hand, there shouldn't be so many references in one single block (e.g., [11-17]) because then it is not possible to assess what is the relevance of the cited literature.
  • Besides the hardware design, comments should be made as to how this  method can be improved by using intelligent systems. Further, more comments on the signal processing side of things would be interesting (smoothening, machine learning, etc.).
  • There should be also further referencing and explanations towards the applicability of the proposes system. For example, such hardware could be used for condition monitoring and prognosis of machines: 'Deep recurrent entropy adaptive model for system reliability monitoring', 'Visually Interpretable Profile Extraction with an Autoencoder for Health Monitoring of Industrial Systems'.
  • Some figures (e.g., 11 and 12) require grids for easier interpretation. In some other figures, the format needs to be improved. For example in figure 17 the subfigures have different sizes.
  • English grammar and style needs to be improved. Some examples: 'Design and Application of MEMS-based Hall Sensor Array to Magnetic Field Mapping' -> 'Design and Application of MEMS-based Hall Sensor Array for Magnetic Field Mapping'?; improve punctuation. 

Author Response

The paper proposes an interesting approach to sensor fusion in magnetic fields, but it requires of a number of amendments prior to publication.

  1. Generic sensor fusion techniques should be referenced. On the other hand, there shouldn't be so many references in one single block (e.g., [11-17]) because then it is not possible to assess what is the relevance of the cited literature.

Authors’ response:

Thanks to the reviewer! Ref [14-17] have been replaced by more relevant literatures.

  1. Besides the hardware design, comments should be made as to how this method can be improved by using intelligent systems. Further, more comments on the signal processing side of things would be interesting (smoothening, machine learning, etc.).

Authors’ response:

Thanks to the reviewer! Line 114-115 and Fig. 3(b) have been added to improve the explanation

of the signal processing.

  1. There should be also further referencing and explanations towards the applicability of the proposes system. For example, such hardware could be used for condition monitoring and prognosis of machines: 'Deep recurrent entropy adaptive model for system reliability monitoring', 'Visually Interpretable Profile Extraction with an Autoencoder for Health Monitoring of Industrial Systems'.

Authors’ response:

Thanks to the reviewer! The two important references have been added in Ref [16-17]

.

  1. Some figures (e.g., 11 and 12) require grids for easier interpretation. In some other figures, the format needs to be improved. For example, in figure 14 the subfigures have different sizes.

Authors’ response:

Thanks to the reviewer! Figs. 11, 12 and 14 have been revised according to the reviewer’s comment.

  1. English grammar and style needs to be improved. Some examples: 'Design and Application of MEMS-based Hall Sensor Array to Magnetic Field Mapping' -> 'Design and Application of MEMS-based Hall Sensor Array for Magnetic Field Mapping'?; improve punctuation. 

Authors’ response:

Thanks to the reviewer! The English grammar of the title and text has been improved according the comment.

Reviewer 2 Report

The article contains original research results.
Authors should improve the quality of the introduction, add more references to works in the area under consideration.
In addition, the quality of Figures 9-10 should be improved. It is necessary to add legends everywhere, increase the font of the text in the figures.

Author Response

The article contains original research results.
1. Authors should improve the quality of the introduction, add more references to works in the area    under consideration.

Authors’ response:

Thanks to the reviewer! Ref [14-17] have been replaced by more relevant literatures.

  1. In addition, the quality of Figures 9-10 should be improved. It is necessary to add legends everywhere, increase the font of the text in the figures.

Authors’ response:

Thanks to the reviewer! Figs. 9 and 10 have been revised according to the reviewer’s comment.

Round 2

Reviewer 1 Report

The paper proposes an interesting topic with an innovative sensor fusion approach. The authors have addressed all the recommendations, and the manuscript is much improved. Hence, I recommend the paper for publication.

Author Response

Thanks to the reviewer.